# Probing Emergent Semantics in Predictive Agents via Question Answering

## Abstract

Recent work has demonstrated how predictive modeling can endow agents with rich knowledge of their surroundings, improving their ability to act in complex environments. We propose question-answering as a general paradigm to decode and understand the representations that such agents develop, applying our method to two recent approaches to predictive modeling – action-conditional CPC (Guo et al., 2018) and SimCore (Gregor et al., 2019). After training agents with these predictive objectives in a visually-rich, 3D environment with an assortment of objects, colors, shapes, and spatial configurations, we probe their internal state representations with a host of synthetic (English) questions, without backpropagating gradients from the question-answering decoder into the agent. The performance of different agents when probed in this way reveals that they learn to encode detailed, and seemingly compositional, information about objects, properties and spatial relations from their physical environment. Our approach is intuitive, *i.e.* humans can easily interpret the responses of the model as opposed to inspecting continuous vectors, and model-agnostic, *i.e.* applicable to any modeling approach. By revealing the implicit knowledge of objects, quantities, properties and relations acquired by agents as they learn, *question-conditional agent probing* can stimulate the design and development of stronger predictive learning objectives.

## 1 Introduction

Some of the biggest successes in artificial intelligence have relied on learning representations from large labeled datasets (Krizhevsky et al., 2012; Sutskever et al., 2014) or dense reward signals (Mnih et al., 2015; Silver et al., 2016). However, an intelligent agent that is capable of functioning in a complex, open-ended environment should be capable of learning general representations that are task-agnostic, and should not require exhaustive labeled data collection or careful reward design.

One of the main challenges in developing such agents is the need for general approaches to evaluate and analyze agents' internal states. In this work, we propose question-answering as an evaluation paradigm for analyzing how much objective knowledge about the external environment is encoded in an agent's internal representation. Our motivation to do so is twofold. First, question-answering provides an intuitive investigative tool for humans – one can simply *ask* an agent what it knows about its environment and get an answer back, without having to inspect internal activations. Second, the space of questions is fairly open-ended – we can pose arbitrarily complex questions to an agent, enabling a comprehensive analysis of its internal states. Question-answering has previously been studied in textual (Rajpurkar et al., 2016; 2018), visual (Malinowski & Fritz, 2014; Antol et al., 2015; Das et al., 2017) and embodied (Gordon et al., 2018; Das et al., 2018a) settings. Crucially, however, these systems are trained end-to-end for the goal of answering questions. Here, we utilize question-answering simply to probe an agent's internal representation, without backpropagating gradients from the question-answering decoder into the agent. That is, we view question-answering as a general purpose decoder of environment knowledge designed to assist the development of agents.

We are particularly interested in agents that can learn general task-agnostic representations of the external world. One promising way to achieve this is via self-supervised predictive modeling. Inspired by learning in humans (Elman, 1990; Quiroga et al., 2005; Nortmann et al., 2013; Rao & Ballard, 1999; Clark, 2016; Hohwy, 2013; Seth, 2015), predictive modeling, *i.e.* predicting future sensory observations, has emerged as a powerful method to learn general-purpose neural network

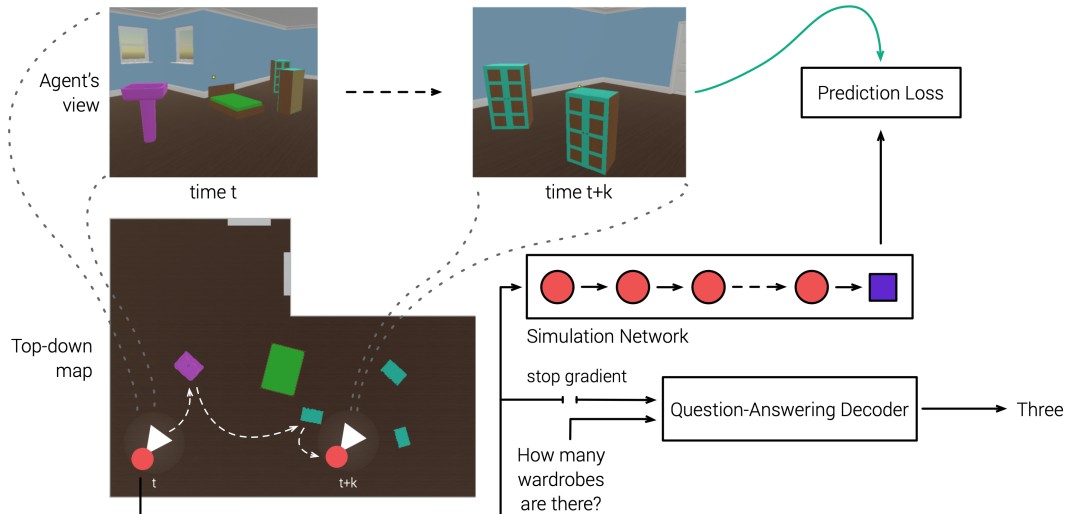

Figure 1: We train predictive agents to explore a visually-rich 3D environment with an assortment of objects of different shapes, colors and sizes. As the agent navigates (trajectory shown in white on the top-down map), it contains an auxiliary network that learns to simulate representations of future observations (labeled 'Simulation Network') say $k$ steps into the future self-supervised by a loss on the agent's future prediction against the ground-truth egocentric observation at $t + k$. Simultaneously, another decoder network is trained to extract answers to a variety of questions about the environment, conditioned on the agent's internal memory but without affecting it (notice 'stop gradient' – gradients from the QA decoder are not backpropagated into the agent). We use this question-answering paradigm to decode and understand the internal representations that such agents develop. Note that the top-down map is only shown for illustration purposes and not available to the agent.

representations (Elias, 1955; Atal & Schroeder, 1970; Schmidhuber, 1991; Schaul & Ring, 2013; Schaul et al., 2015; Silver et al., 2017; Wayne et al., 2018; Guo et al., 2018; Gregor et al., 2019; Recanatesi et al., 2019). These representations can be learned while exploring in and interacting with an environment in a task-agnostic manner, and later exploited for goal-directed behavior.

We evaluate predictive *vs.* non-predictive agents (both trained for exploration) on our question-answering testbed to investigate how much objective knowledge about environment semantics can be captured *solely by egocentric prediction*. By *semantics*, here we specifically refer to information about objects – quantity, colors, shapes, spatial relations. The set of questions is intended to be holistic, *i.e.* they require a representation of relevant aspects of the whole environment and in general cannot be answered from a single observation, nor a few consecutive observations of the episode – and test a variety of local and global scene understanding, visual reasoning, and recall skills.

Concretely, we make the following contributions:

- In a visually rich 3D room environment developed in the Unity game engine[1], we define and develop a set of questions designed to probe a range of semantic, relational and spatial knowledge – from identifying shapes and colors ('What shape is the red object?') to counting ('How many blue objects are there?') to spatial relations ('What is the color of the chair near the table?'), exhaustive search ('Is there a cushion?'), and comparisons ('Are there the same number of tables as chairs?').

- We train RL agents augmented with predictive loss functions – 1) action-conditional CPC (Guo et al., 2018) and 2) SimCore (Gregor et al., 2019) – for an exploration task and analyze the internal representations they develop by decoding answers to our suite of questions. Crucially, the QA decoder is trained independent of the predictive agent and we find that QA performance is indicative of the agent's ability to capture global environment structure and semantics *solely through egocentric prediction*. We compare these predictive agents to strong non-predictive LSTM baselines as well as to an agent that is explicitly optimized for the question-answering task.

- We establish generality of the semantic knowledge by testing zero-shot generalization of a trained QA decoder to compositionally novel questions (unseen combinations of seen attributes), suggesting a degree of compositionality in the internal representations captured by predictive agents.

---

[1] unity3d.com

## 2 RELATED WORK

Our work relates to, and builds on, prior work on predictive modeling and auxiliary loss functions in reinforcement learning as well as grounded language learning and embodied question answering.

**Predictive modeling and auxiliary loss functions in RL**. The power of predictive modeling for representation learning has been known since at least the seminal work of Elman (1990) on emergent language structures. More recent examples include Word2Vec (Mikolov et al., 2013), Skip-Thought vectors (Kiros et al., 2015), and BERT (Devlin et al., 2019) in language, while in vision similar principles have been applied to context prediction (Doersch et al., 2015; Noroozi & Favaro, 2016), unsupervised tracking (Wang & Gupta, 2015), inpainting (Pathak et al., 2016) and colorization (Zhang et al., 2016). More related to us is the use of such techniques in designing auxiliary loss functions for training model-free RL agents, such as successor representations (Dayan, 1993; Zhu et al., 2017a), value and reward prediction (Jaderberg et al., 2016; Wayne et al., 2018), contrastive predictive coding (CPC) (Oord et al., 2018; Hénaff et al., 2019), and SimCore (Gregor et al., 2019).

**Grounded language learning**. Inspired by the work of Winograd (1972) on SHRDLU, several recent works have explored linguistic representation learning by grounding language into actions and pixels in physical environments – in 2D gridworlds (Andreas et al., 2017; Yu et al., 2018; Misra et al., 2017), 3D (Chaplot et al., 2018; Hermann et al., 2017; Das et al., 2018a; Gordon et al., 2018; Cangea et al., 2019; Puig et al., 2018; Zhu et al., 2017a; Anderson et al., 2018; Gupta et al., 2017; Zhu et al., 2017b; Oh et al., 2017; Shu et al., 2018; Vogel & Jurafsky, 2010) and textual (Matuszek et al., 2013; Narasimhan et al., 2015) environments. Closest to our work is the task of Embodied Question Answering (Gordon et al., 2018; Das et al., 2018a;b; Yu et al., 2019; Wijmans et al., 2019) – where an embodied agent in an environment (*e.g.* a house) is asked to answer a question (*e.g.* "What color is the piano?"). Typical approaches to EmbodiedQA involve training agents to move for the goal of answering questions. In contrast, our focus is on learning a good predictive model in a *goal-agnostic* exploration phase and using question-answering as a post-hoc testbed for evaluating the semantic knowledge that can emerge in the agent's representations from predicting the future.

**Neural population decoding**. Probing an agent using a QA decoder can be viewed as a variant of neural population decoding, successfully used as an analysis tool in neuroscience (Georgopoulos et al., 1986; Bialek et al., 1991; Salinas & Abbott, 1994) and more recently in deep learning (Guo et al., 2018; Gregor et al., 2019; Azar et al., 2019; Alain & Bengio, 2016). The idea is to test if desired information is encoded in the learned representation by feeding it as input to a network tasked to extract the desired information. In deep learning, this is done by training a network to predict the desired parts of the ground-truth state of the environment, such as an agent's position or orientation, without backpropagating through the agent's internal state. We extend this idea by using question-answering as a general purpose decoder conditioned by an arbitrary question, through which we can attempt to extract complex high-level information from the agent's internal state.

**Neurocognition**. Predictive modeling is thought to be a fundamental component of human neurocognition (Elman, 1990; Clark, 2016; Hohwy, 2013; Seth, 2015). In particular, it has been proposed that human learning and decision-making rely on the minimization of prediction error (Clark, 2016; Friston, 2010; Friston et al., 2017; Hohwy, 2013; Seth, 2015). A well-established strand of work has focused on decoding predictive representations in brain states (Quiroga et al., 2005; Nortmann et al., 2013; Huth et al., 2016). The question of how prediction of sensory experience relates to higher-order conceptual knowledge is complex and subject to debate (Williams, 2018; Roskies & Wood, 2017), though some have proposed that conceptual knowledge, planning, reasoning, and other higher-order functions emerge in deeper layers of a predictive network. We focus on the emergence of semantics in an artificial predictive agent's internal representations.

## 3 ENVIRONMENT & TASKS

**Environment.** We use a Unity-based visually-rich 3D environment (see Figure 1). It is a single L-shaped room that can be programmatically populated with an assortment of objects of different colors at different spatial locations and orientations. In total, we use a library of 50 different objects, referred to as 'shapes' henceforth (*e.g.* chair, teddy, glass, *etc.*), in 10 different colors (*e.g.* red, blue, green, *etc.*). For a complete list of objects, attributes, and other environment details, see section A.4.

At every step, the agent gets a $96 \times 72$ first-person RGB image as its observation, and the action space consists of movements (`move-{forward,back,left,right}`), turns (`turn-{up,down,left,right}`), and object pick-up and manipulation (4 DoF: yaw, pitch, roll, and movement along the axis between the agent and object). See Table 5 in the Appendix for the full set of actions.

| Question type | Template | Level codename | # QA pairs |
|---|---|---|---|
| Attribute | What is the color of the <shape>? | `color` | 500 |
| | What shape is the <color> object? | `shape` | 500 |
| Count | How many <shape> are there? | `count_shape` | 200 |
| | How many <color> objects are there? | `count_color` | 40 |
| Exist | Is there a <shape>? | `existence_shape` | 100 |
| Compare + Count | Are there the same number of <color1> objects as <color2> objects? | `compare_count_color` | 180 |
| | Are there the same number of <shape1> as <shape2>? | `compare_count_shape` | 4900 |
| Relation + Attribute | What is the color of the <shape1> near the <shape2>? | `near_color` | 24500 |
| | What is the <color> object near the <shape>? | `near_shape` | 25000 |

Table 1: Question-answering task templates. In every episode, objects and their configurations are randomly generated, and these task templates get translated to question-answer pairs for all unambiguous <shape, color> combinations. There are 50 shapes and 10 colors in total. For more details, see section A.4.

**Question-Answering Tasks.** We develop a range of question-answering tasks of varying complexity that test the agent's local and global scene understanding, visual reasoning, and memory skills. Inspired by Johnson et al. (2017); Das et al. (2018a); Gordon et al. (2018), we programmatically generate a dataset of questions (see Table 1). These questions ask about the presence or absence of objects (`existence_shape`), their attributes (`color`, `shape`), counts (`count_color`, `count_shape`), quantitative comparisons (`compare_count_color`, `compare_count_shape`), and elementary spatial relations (`near_color`, `near_shape`). Unlike the fully-observable setting in CLEVR (Johnson et al., 2017), the agent does not get a global view of the environment, and must answer these questions from a sequence of partial egocentric observations. Moreover, unlike prior work on Embodied Question Answering (Gordon et al., 2018; Das et al., 2018a), the agent is *not* being trained end-to-end to move to answer questions. It is being trained to explore, and answers are being decoded (without backpropagating gradients) from its internal representation. Thus in order to answer these questions, the agent *must* learn to encode relevant aspects of the environment in a representation amenable to easy decoding into symbols (*e.g.* what does the word "chair" mean? or what representations does computing "how many" require?).

## 4 APPROACH

**Learning an exploration policy**. Predictive modeling has proven to be an effective approach for an agent to develop general knowledge of its environment as it explores and behaves in the service of its principal goal, typically maximising environment returns (Gregor et al., 2019; Guo et al., 2018). Since we wish to evaluate the effectiveness of predictive modeling independent of the agent's specific goal, we define a simple task that stimulates the agent to visit all of the 'important' places in the environment (*i.e.* to acquire an exploratory but otherwise task-neutral policy). This is achieved by giving the agent a reward of $+1.0$ every time it visits an object in the room for the first time. After visiting all objects, rewards are refreshed and available to be consumed by the agent again (*i.e.* re-visiting an object the agent has already been to will now again lead to a $+1.0$ reward), and this process continues for the duration of each episode (30 seconds or 900 steps).

During training on this exploration task, the agent receives a first-person RGB observation $x_t$ at every timestep $t$, and processes it using a convolutional neural network to produce $z_t$. This is input to an LSTM policy whose hidden state is $h_t$ and output a discrete action $a_t$. The agent optimizes the discounted sum of future rewards using an importance-weighted actor-critic algorithm (Espeholt et al., 2018).

**Training the QA-decoder**. The question-answering decoder is operationalized as an LSTM that is initialized with the agent's internal representation $h_t$ and receives the question as input at every timestep (see Figure 2). The question is a string that we tokenise into words and then map to learned embeddings. The question decoder LSTM is then unrolled for a fixed number of computation steps after which it predicts a softmax distribution over the vocabulary of one-word answers to questions

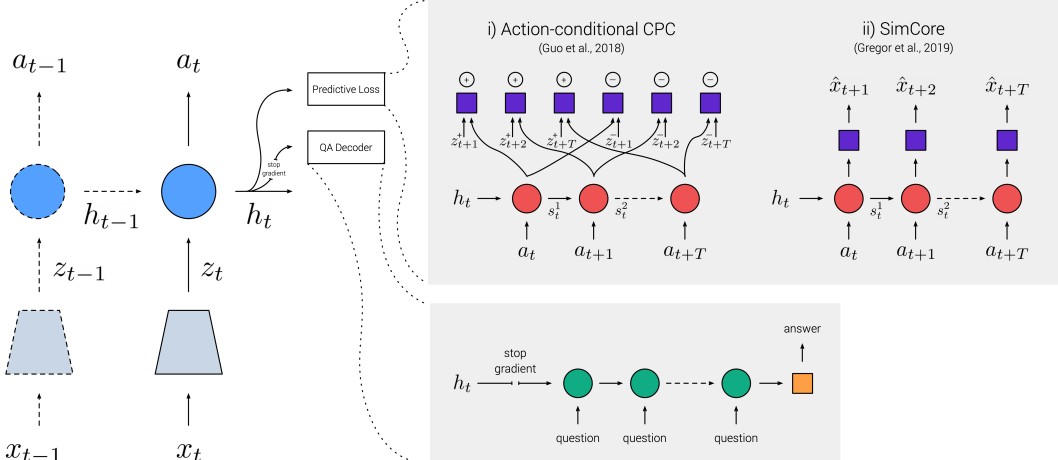

Figure 2: Overview of our approach: at every timestep $t$, the agent receives an RGB observation $x_t$ as input, processes it using a convolutional neural network to produce $z_t$, which is then processed by an LSTM to select action $a_t$. The agent is learning to explore – it receives a reward of 1.0 for navigating to each object in the environment. As it explores the environment, it builds up an internal representation $h_t$, which receives pressure from an auxiliary predictive module to capture environment semantics so as to accurately predict consequences of its actions multiple steps into the future. We experiment with a vanilla LSTM agent and two recent predictive approaches – action-conditional CPC (Guo et al., 2018) and SimCore (Gregor et al., 2019). The learnt internal representations are then analyzed via a question-answering decoder whose gradients are not backpropagated to the agent core. The QA decoder is an LSTM initialized with $h_t$ and receiving the question at every timestep.

in Table 1, and is trained via a cross-entropy loss. Crucially, this QA decoder is trained independent of the agent policy; *i.e.* gradients from this decoder are not allowed to flow back into the agent. We evaluate question-answering performance by measuring top-1 accuracy at the end of the episode – we consider the agent's top predicted answer at the last time step of the episode and compare that with the ground-truth answer.

The QA decoder can be seen as a general purpose decoder trained to extract object-specific knowledge from the agent's internal state without affecting the agent itself. If this knowledge is not retained in the agent's internal state, then this decoder will not be able to extract it. This is an important difference with respect to prior work (Gordon et al., 2018; Das et al., 2018a) – wherein agents were trained to move to answer questions, *i.e.* all parameters had access to linguistic information. Recall that the agent's navigation policy has been trained for exploration, and so the visual information required to answer a question need not be present in the observation at the end of the episode. Thus, through question-answering, we are evaluating the degree to which agents encode relevant aspects of the environment (object colors, shapes, counts, spatial relations) in their internal representations *and* maintain this information in memory beyond the point at which it was initially received[2].

## 4.1 Auxiliary Predictive Losses

We augment the baseline architecture described above with an auxiliary predictive head consisting of a simulation network (operationalized as an LSTM) that is initialized with the agent's internal state $h_t$ and deterministically simulates future latent states $s_t^1, \ldots, s_t^k, \ldots$ in an open-loop manner, receiving the agent's action sequence as input. We evaluate two predictive losses – action-conditional CPC (Guo et al., 2018) and SimCore (Gregor et al., 2019). See Fig. 2 for overview, A.1.2 for details.

**Action-conditional CPC** (CPC|A, Guo et al. (2018)) makes use of a noise contrastive estimation model to discriminate between true observations processed by the convolutional neural network $z_{t+k}^+$ ($k$ steps into the future) and negatives randomly sampled from the dataset $z_{t+k}^-$, in our case from other episodes in the minibatch. Specifically, at each timestep $t + k$ (up to a maximum), the output of the simulation core $s_t^k$ and $z_{t+k}^+$ are fed to an MLP to predict 1, and $s_t^k$ and $z_{t+k}^-$ are used to predict 0.

---

[2]See Section A.1.3 for more details about the QA decoder.

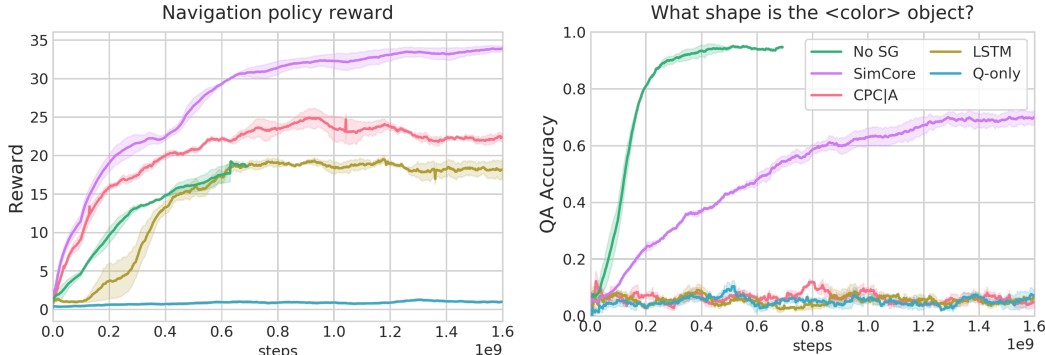

Figure 3: L – Reward in an episode. R – Top-1 QA accuracy. Averaged over 3 seeds. Shaded region is 1 SD.

**SimCore** (Gregor et al., 2019) uses the simulated state $s_t^k$ to condition a generative model based on ConvDRAW (Gregor et al., 2016) and GECO (Rezende & Viola, 2018) that predicts the distribution of true observations $p(x_{t+k}|h_t, a_{t,...,(t+k)})$ in pixel space.

**Baselines**. We evaluate and compare the above approaches with 1) a vanilla RL agent without any auxiliary predictive losses (referred to as 'LSTM'), and 2) a question-only agent that receives zero-masked observations as input and is useful to measure biases in our question-answering testbed. Such a baseline is critical, particularly when working with simulated environments, as it can uncover biases in the environment's generation of tasks that can result in strong but uninteresting performance from agents capable of powerful function approximation (Thomason et al., 2019).

**No stop gradient**. We also compare against an agent without blocking the QA decoder gradients (labeled 'No SG'). This model differs from the above in that it is trained end-to-end – with supervision – to answer the set of questions in addition to the exploration task. Hence, it represents an agent receiving privileged information about how to answer the questions and its performance provides an upper bound indicative of how challenging these question-answering tasks are in this context.

## 5 EXPERIMENTS & RESULTS

### 5.1 QUESTION-ANSWERING PERFORMANCE

We begin by analyzing performance on a single question-answering task – shape – which consists of questions of the form "what shape is the <color> object?". Figure 3 shows the average reward accumulated by the agent in one episode (left) and the QA accuracy at the last timestep of the episode (right) for all approaches over the course of training. We make the following observations:

- **All agents learn to explore**. With the exception 'question-only', all agents achieve high reward on the exploration task. This means that they visited all objects in the room more than once each and therefore, in principle, have been exposed to sufficient information to answer all questions.

- **Predictive models aid navigation**. Agents equipped with auxiliary predictive losses – CPC|A and SimCore – collect the most rewards, suggesting that predictive modeling helps navigate the environment efficiently. This is consistent with findings in Gregor et al. (2019).

- **QA decoding from LSTM and CPC|A representations is no better than chance**.

- **SimCore's internal representations lead to best QA accuracy**. SimCore gets to a QA accuracy of ~72% indicating that its representations best capture environment knowledge and are the most suitable for decoding answers to questions. Figure 4a shows example agent predictions.

- **Wide gap between SimCore and No SG**. There is still a ~24% gap between SimCore and the No SG oracle, suggesting scope for developing better auxiliary predictive losses.

It is worth emphasizing that answering this shape question from observations is not a challenging task in and of itself. The No SG agent, which is trained end-to-end to optimize both for exploration and QA, achieves almost-perfect accuracy (~96%). The challenge arises from the fact that we are not training the agent end-to-end – from pixels to navigation to QA – but decoding the answer from

| | Overall | shape | color | exist | count_shape | count_color | compare_count_color | compare_count_shape | near_shape | near_color |
|---|---|---|---|---|---|---|---|---|---|---|
| Baseline: Question-only | 0.29 | 0.04 | 0.1 | 0.63 | 0.24 | 0.24 | 0.49 | 0.70 | 0.04 | 0.09 |
| LSTM | 0.31 | 0.04 | 0.1 | 0.54 | 0.34 | 0.38 | 0.53 | 0.70 | 0.04 | 0.09 |
| CPC\|A | 0.32 | 0.06 | 0.08 | 0.64 | **0.39** | 0.39 | 0.50 | 0.70 | 0.06 | 0.10 |
| SimCore | **0.60** | **0.72** | **0.81** | **0.72** | **0.39** | **0.57** | **0.56** | **0.73** | **0.30** | **0.59** |
| Oracle: No SG | 0.63 | 0.96 | 0.81 | 0.60 | 0.45 | 0.57 | 0.51 | 0.76 | 0.41 | 0.72 |

Table 2: Top-1 accuracy on question-answering tasks.

the agent's internal state, which is learned agnostic to the question. The answer can only be decoded if the agent's internal state contains relevant information represented in an easily-decodable way.

**Decoder complexity**. To explore the possibility that answer-relevant information is present in the agent's internal state but requires a more powerful decoder, we experiment with QA decoders of a range of depths. As detailed in Figure 6 in the appendix, we find that using a deeper QA decoder with SimCore does lead to higher QA accuracy (from $1 \rightarrow 12$ layers), although greater decoder depths become detrimental after 12 layers. Crucially, however, in the non-predictive LSTM agent, the correct answer cannot be decoded irrespective of the capacity of the QA decoder. This highlights an important aspect of our question-answering evaluation paradigm – that while the absolute accuracy at answering questions may also depend on decoder capacity, relative differences provide an informative comparison between the internal representations developed by different agents.

Table 2 shows QA accuracy for all QA tasks (see Figure 7 in appendix for training curves). The results reveal large variability in difficulty across question types. Questions about attributes (color and shape), which can be answered from a single well-chosen frame of visual experience, are the easiest, followed by spatial relationship questions (near_color and near_shape), and the hardest are counting questions (count_color and count_shape). We further note that:

- **All agents perform $\geqslant$ question-only**, which captures any biases in the environment or question distributions (enabling strategies such as constant prediction of the most-common answer).
- **CPC\|A representations are not better than LSTM on most question types**.
- **SimCore representations achieve higher QA accuracy than all other approaches**, substantially above question-only on count_color ($57\%$ *vs.* $24\%$), near_shape ($30\%$ *vs.* $4\%$) and near_color ($59\%$ *vs.* $9\%$), demonstrating a strong tendency for encoding and retaining information about the identity of objects, their properties, and both spatial as well as temporal relations.

Finally, as before, the No SG agent trained to answer questions without stopped gradients achieves highest accuracy for most questions, although not all – perhaps due to trade-offs between simultaneously optimizing performance for different QA losses and the exploration task.

## 5.2 COMPOSITIONAL GENERALIZATION

While there is a high degree of procedural randomization in our environment and QA tasks, over-parameterized neural-network-based models in limited environments are always prone to overfitting or rote memorization. We therefore constructed a test of the generality of the information encoded in the internal state of an agent. The test involves a variant of the shape question type (*i.e.* questions like "what shape is the <color> object?"), but in which the possible question-answer pairs are partitioned into mutually exclusive training and test splits. Specifically, the test questions are constrained such that they are compositionally novel – the <color, shape> combination involved in the question-answer pair is never observed during training, but both attributes are observed in other contexts. For instance, a test question-answer pair "Q: what shape is the **blue** object?, A: **table**" is excluded from the training set of the QA decoder, but "Q: what shape is the **blue** object?, A: **car**" and "Q: What shape is the **green** object?, A: **table**" are part of the training set (but not the test set).

We evaluate only the SimCore agent on this test of generalization (since other agents perform poorly on the original task itself). Figure 4b shows that the QA decoder applied to SimCore's internal states performs at substantially above-chance (and all baselines) on the held-out test questions (although somewhat lower than training performance). This indicates that the QA decoder extracts and applies

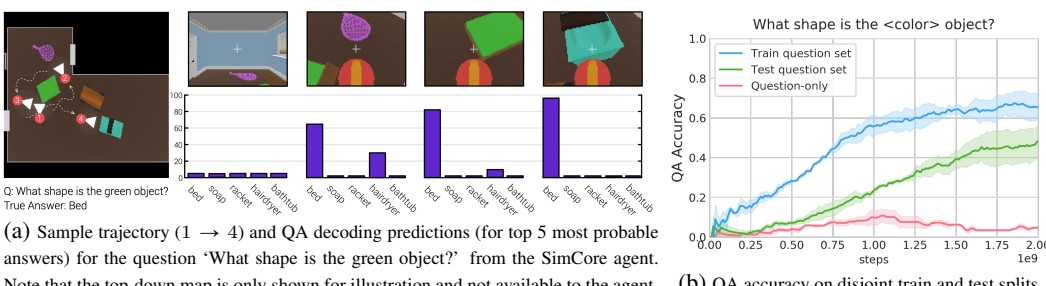

(a) Sample trajectory (1 → 4) and QA decoding predictions (for top 5 most probable answers) for the question 'What shape is the green object?' from the SimCore agent. Note that the top-down map is only shown for illustration and not available to the agent.

(b) QA accuracy on disjoint train and test splits.

Figure 4

information in a comparatively factorized (or compositional) manner, and suggests (circumstantially) that the knowledge acquired by the simcore may also be represented in this way.

## 6 DISCUSSION

We introduced question-answering as a paradigm to evaluate and analyze representations learned by artificial agents. In particular, we tested how much knowledge about the external environment can be decoded from predictive *vs.* non-predictive RL agents. We started by developing a range of question-answering tasks in a visually-rich 3D environment serving as a diagnostic test of an agent's scene understanding, visual reasoning, and memory skills. Next, we trained agents to optimize an exploration objective with and without auxiliary self-supervised predictive losses, and evaluated the representations they form as they explore an environment via this question-answering testbed. We found that predictive agents (in particular SimCore (Gregor et al., 2019)) are able to reliably capture detailed environment semantics in their internal states, which can be easily decoded as answers to questions, while non-predictive agents do not, even if they optimize the exploration objective well.

Interestingly, not all predictive agents are equally good at forming these representations. We compared a model explicitly learning the probability distribution of future frames in pixel space via a generative model (SimCore (Gregor et al., 2019)) with a model based on discriminating frames through contrastive estimation (CPC|A (Guo et al., 2018)). We found that while both learned to navigate well, only the former developed representations that could be used for answering questions about the environment. Gregor et al. (2019) previously showed that the choice of predictive model has a significant impact on the ability to decode an agent's position, orientation and top-down map reconstructions of the environment. Here we extend this idea to more high-level and complex aspects of the environment and show the value of our question-answering approach in comparing existing agents and its potential utility as a tool for developing better ones.

Finally, the fact that we can even decode answers to questions (*i.e.* symbolic information) from an agent's internal representations learned solely from egocentric future predictions without exposing the agent to *any* questions is encouraging. It indicates that the agent is learning to form and maintain invariant object identities and properties (modulo limitations in decoder capacity) in its internal state *without explicit supervision*. It is almost 30 years since Elman (1990) showed how syntactic structures and semantic organization can emerge in the units of a neural network as a consequence of the simple objective of predicting the next word in a sequence. This work corroborates Elman's belief in the power of prediction by demonstrating the diversity of knowledge that can emerge when a situated neural-network agent is endowed with powerful predictive objectives applied to raw pixel observations. We think we have just scratched the surface of this problem, and hope our work inspires future research in evaluating predictive agents using natural linguistic interactions.

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

## A   APPENDIX

### A.1   NETWORK ARCHITECTURES AND TRAINING SETUP

#### A.1.1   IMPORTANCE WEIGHTED ACTOR-LEARNER ARCHITECTURE

Agents were trained using the IMPALA framework (Espeholt et al., 2018). Briefly, there are N parallel 'actors' collecting experience from the environment in a replay buffer and one learner taking batches of trajectories and performing the learning updates. During one learning update the agent network is unrolled, all the losses (RL and auxiliary ones) are evaluated and the gradients computed.

#### A.1.2   AGENTS

**Input encoder** To process the frame input, all models in this work use a residual network (He et al., 2016) of 6 64-channel ResNet blocks with rectified linear activation functions and bottleneck channel of size 32. We use strides of (2, 1, 2, 1, 2, 1) and don't use batch-norm. Following the convnet we flatten the ouput and use a linear layer to reduce the size to 500 dimensions. Finally, We concatenate this encoding of the frame together with a one hot encoding of the previous action and the previous reward.

**Core architecture** The recurrent core of all agents is a 2-layer LSTM with 256 hidden units per layer. At each time step this core consumes the input embedding described above and update its state. We then use a 200 units single layer MLP to compute a value baseline and an equivalent network to compute action logits, from where one discrete action is sampled.

**Simulation Network** Both predictive agents have a simulation network with the same architecture as the agent's core. This network is initialized with the agent state at some random time $t$ from the trajectory and unrolled forward for a random number of steps up to 16, receiving only the actions of the agent as inputs. We then use the resulting LSTM hidden state as conditional input for the prediction loss (SimCore or CPC|A).

**SimCore** We use the same architecture and hyperparameters described in Gregor et al. (2019). The output of the simulation network is used to condition a Convolutional DRAW (Gregor et al., 2016). This is a conditional deep variational auto-encoder with recurrent encoder and decoder using convolutional operations and a canvas that accumulates the results at each step to compute the distribution over inputs. It features a recurrent prior network that receives the conditioning vector and computes a prior over the latent variables. See more details in Gregor et al. (2019).

**Action-conditional CPC** We replicate the architecture used in Guo et al. (2018). CPC|A uses the output of the simulation network as input to an MLP that is trained to discriminate true versus false future frame embedding. Specifically, the simulation network outputs a conditioning vector after $k$ simulation steps which is concatenated with the frame embedding $z_{t+k}$ produced by the image encoder on the frame $x_{t+k}$ and sent through the MLP discriminator. The discriminator has one hidden layer of 512 units, ReLU activations and a linear output of size 1 which is trained to binary classify true embeddings into one class and false embeddings into another. We take the negative examples from random time points in the same batch of trajectories.

### A.1.3   QA NETWORK ARCHITECTURE

**Question encoding** The question string is first tokenized to words and then mapped to integers corresponding to vocabulary indices. These are then used to lookup 32-dimensional embeddings for each word. We then unroll a 64-units single-layer LSTM for a fixed number of 15 steps. The language representation is then computed by summing the hidden states for all time steps.

**QA decoder**. To decode answers from the internal state of the agents we use a second LSTM initialized with the internal state of the agent's LSTM and unroll it for a fix number of steps, consuming the question embedding at each step. The results reported in the main section were computed using 12 decoding steps. The terminal state is sent through a two-layer MLP (sizes 256, 256) to compute a vector of answer logits with the size of the vocabulary and output the top-1 answer.

### A.1.4   HYPER-PARAMETERS

The hyper-parameter values used in all the experiments are in Table 3.

| Agent | |
|---|---|
| Learning rate | 1e-4 |
| Unroll length | 50 |
| Adam $\beta_1$ | 0.90 |
| Adam $\beta_2$ | 0.95 |
| Policy entropy regularization | 0.0003 |
| Discount factor | 0.99 |
| No. of ResNet blocks | 6 |
| No. of channel in ResNet block | 64 |
| Frame embedding size | 500 |
| No. of LSTM layers | 2 |
| No. of units per LSTM layer | 256 |
| No. of units in value MLP | 200 |
| No. of units in policy MLP | 200 |
| **Simulation Network** | |
| Overshoot length | 16 |
| No. of LSTM layers | 2 |
| No. of units per LSTM layer | 256 |
| No. of simulations per trajectory | 6 |
| No. of evaluations per overshoot | 2 |
| **SimCore** | |
| No. of ConvDRAW Steps | 8 |
| GECO kappa | 0.0015 |
| **CPC\|A** | |
| MLP discriminator size | 64 |
| **QA network** | |
| Vocabulary size | 1000 |
| Maximum question length | 15 |
| No. of units in Text LSTM encoder | 64 |
| Question embedding size | 32 |
| No. of LSTM layers in question decoder | 2 |
| No. of units per LSTM layer | 256 |
| No. of units in question decoder MLP | 200 |
| No. of decoding steps | 12 |

Table 3: Hyperparameters.

### A.1.5 NEGATIVE SAMPLING STRATEGIES FOR CPC|A

We experimented with multiple sampling strategies for the CPC|A agent (whether or not negative examples are sampled from the same trajectory, the number of contrastive prediction steps, the number of negative examples). We report the best results in the main text. The CPC|A agent did provide better representations of the environment than the LSTM-based agent, as shown by the top-down view reconstruction loss (Figure 5a). However, none of the CPC|A agent variations that we tried led to better-than-chance question-answering accuracy. As an example, in Figure 5b we compare sampling negatives from the same trajectory or from any trajectory in the training batch.

### A.2 EFFECT OF QA NETWORK DEPTH

To study the effect of the QA network capacity on the answer accuracy, we tested decoders of different depths applied to both the SimCore and the LSTM agent's internal representations (6). The QA network is an LSTM initialized with the agent's internal state that we unroll for a fix number of steps feeding the question as input at each step. We found that, indeed, the answering accuracy increased with the number of unroll steps from 1 to 12, while greater number of steps became detrimental. We performed the same analysis on the LSTM agent and found that regardless of the capacity of the QA network, we could not decode the correct answer from its internal state, suggesting that the limiting

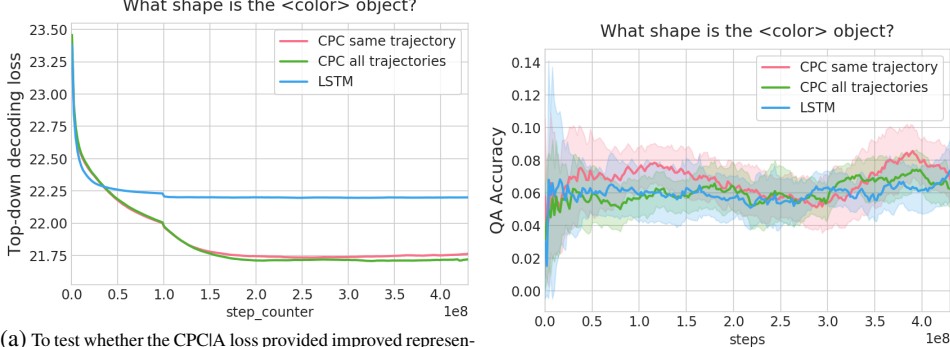

(a) To test whether the CPC|A loss provided improved representations we reconstructed the environment top-down view, similar to Gregor et al. (2019). Indeed the reconstruction loss is lower for CPC|A than for the LSTM agent.

(b) QA accuracy for the CPC|A agent is not better than the LSTM agent, for both sampling strategies of negatives.

Figure 5

factor is not the capacity of the decoder but the lack of useful representations in the LSTM agent state.

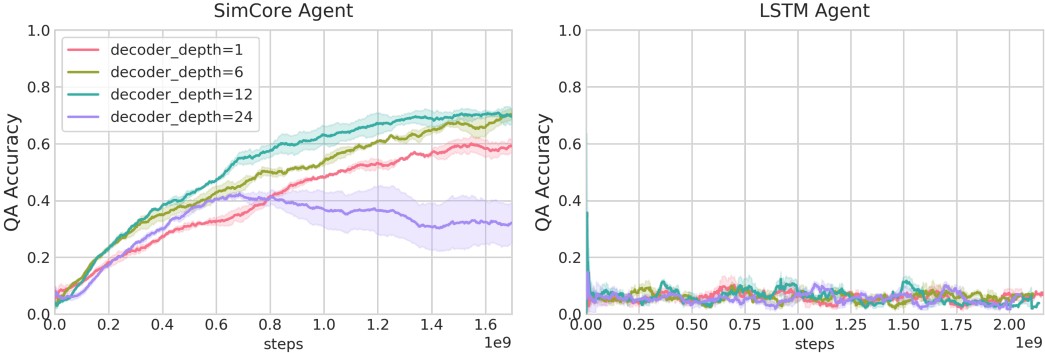

Figure 6: Answer accuracy over training for increasing QA decoder's depths. Left subplot shows the results for the SimCore agent and right subplot for the LSTM baseline. For SimCore, the QA accuracy increases with the decoder depth, up to 12 layers. For the LSTM agent, QA accuracy is not better than chance regardless of the capacity of the QA network.

### A.3 ANSWERING ACCURACY DURING TRAINING FOR ALL QUESTIONS

The QA accuracy over training for all questions is shown in Figure 7.

### A.4 ENVIRONMENT

Our environment is a single L-shaped 3D room, procedurally populated with an assortment of objects.

**Actions and Observations.** The environment is episodic, and runs at 30 frames per second. Each episode takes 30 seconds (or 900 steps). At each step, the environment provides the agent with two observations: a 96x72 RGB image with the first-person view of the agent and the text containing the question.

The agent can interact with the environment by providing multiple simultaneous actions to control movement (forward/back, left/right), looking (up/down, left/right), picking up and manipulating objects (4 degrees of freedom: yaw, pitch, roll + movement along the axis between agent and object).

**Rewards.** To allow training using cross-entropy, as described in Section 4, the environment provides the ground-truth answer instead of the reward to the agent.

**Object creation and placement.** We generate between 2 and 20 objects, depending on the task, with the type of the object, its color and size being uniformly sampled from the set described in Table 4.

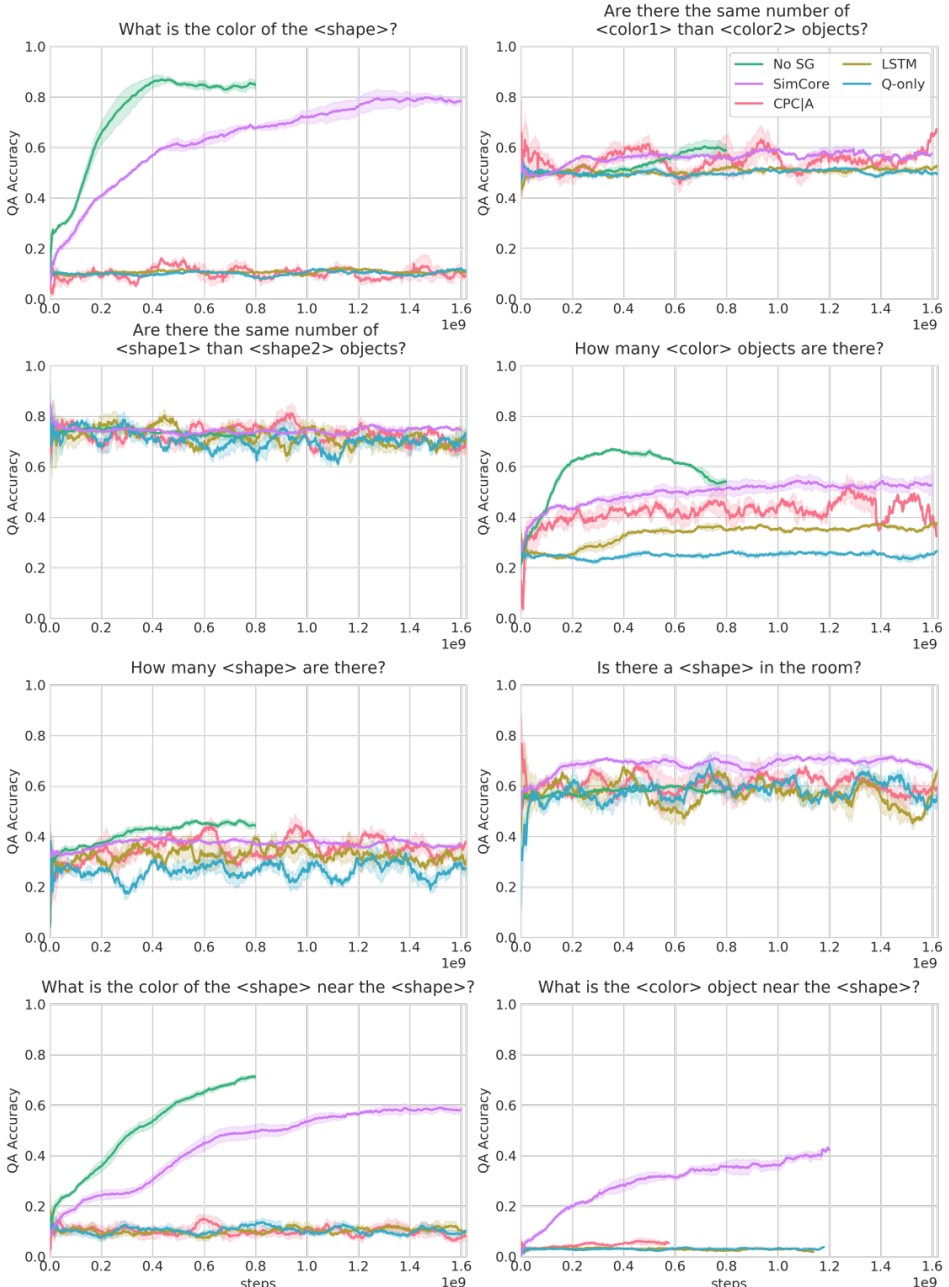

Figure 7: QA accuracy over training for all questions and all models.

Objects will be placed in a random location and random orientation. For some tasks, we required some additional constraints - for example, if the question is "What is the color of the cushion near the bed?", we need to ensure only one cushion is close to the bed. This was done by checking the constraints and regenerating the placement in case they were not satisfied.

## A.5 RESULTS IN THE DEEPMIND LAB ENVIRONMENT

In order to check if our results are robust to the choice of environment, we developed a similar setup using the DeepMind Lab (Beattie et al., 2016) environment. It consists of a rectangular room that is populated with a random selection of objects of different shapes and colors in each episode. There are 6 distinct objects in each room, selected from a pool of 20 objects and 9 different colors. We use a similar exploration reward structure as in our earlier environment to train the agents to navigate

| Attribute | Options |
|---|---|
| Object | basketball, cushion, carriage, train, grinder, candle, teddy, chair, scissors, stool, book, football, rubber duck, glass, toothpaste, arm chair, robot, hairdryer, cube block, bathtub, TV, plane, cuboid block, car, tv cabinet, plate, soap, rocket, dining table, pillar block, potted plant, boat, tennisball, tape dispenser, pencil, wash basin, vase, picture frame, bottle, bed, helicopter, napkin, table lamp, wardrobe, racket, keyboard, chest, bus, roof block, toilet |
| Color | aquamarine, blue, green, magenta, orange, purple, pink, red, white, yellow |
| Size | small, medium, large |

Table 4: Randomization of objects in the Unity room. 50 different types, 10 different colors and 3 different scales.

| Body movement actions | Movement and grip actions | Object manipulation |
|---|---|---|
| NOOP | GRAB | GRAB + SPIN_OBJECT_RIGHT |
| MOVE_FORWARD | GRAB + MOVE_FORWARD | GRAB + SPIN_OBJECT_LEFT |
| MOVE_BACKWARD | GRAB + MOVE_BACKWARD | GRAB + SPIN_OBJECT_UP |
| MOVE_RIGHT | GRAB + MOVE_RIGHT | GRAB + SPIN_OBJECT_DOWN |
| MOVE_LEFT | GRAB + MOVE_BACKWARD | GRAB + SPIN_OBJECT_FORWARD |
| LOOK_RIGHT | GRAB + LOOK_RIGHT | GRAB + SPIN_OBJECT_BACKWARD |
| LOOK_LEFT | GRAB + LOOK_LEFT | GRAB + PUSH_OBJECT_AWAY |
| LOOK_UP | GRAB + LOOK_UP | GRAB + PULL_OBJECT_CLOSE |
| LOOK_DOWN | GRAB + LOOK_DOWN | |

Table 5: Environment action set.

and observe all objects. Finally, in each episode, we introduce a question of the form 'What is the color of the <shape>?' where <shape> is replaced by the name of an object present in the room.

Figure 8b shows question-answering accuracies in the DeepMind Lab environment. Consistent with the results presented in the main text, internal representations of the SimCore agent lead to the highest answering accuracy while CPC|A and the vanilla LSTM agent perform worse and similar to each other. Crucially, for running experiments in DeepMind Lab, we *did not* change any hyperparameters from the experimental setup described in the main paper. This demonstrates that our approach is not specific to a single environment and that it can be readily applied in a variety of settings.

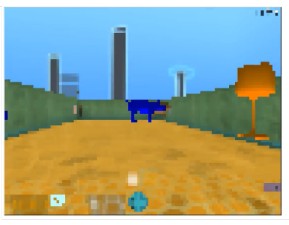 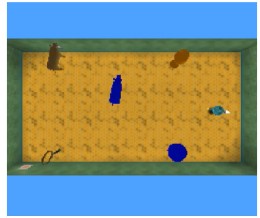

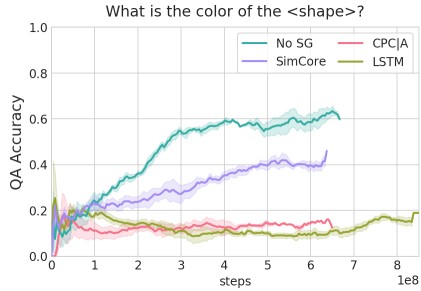

(a) DeepMind Lab environment (Beattie et al., 2016): Rectangular-shaped room with 6 randomly selected objects out of a pool of 20 different objects of different colors.

(b) QA accuracy for `color` questions (What is the color of the <shape>?) in Deep-Mind Lab. Consistent with results in the main paper, internal representations of the SimCore agent lead to the highest accuracy while CPC|A and LSTM perform worse and similar to each other.

Figure 8

