# OpenReview forum: "Probing Emergent Semantics in Predictive Agents via Question Answering"
_ICLR.cc/2020/Conference — Reject_

### Official Review · AnonReviewer2 · 2019-10-23
**Official Blind Review #2**

**Rating:** 8

**Review:**

The authors propose question answering (QA) as a tool to investigate what agents learn about the world, i.e., how much about the world is encoded in their internal states. The authors argue that this is an intuitive method for humans and allows for arbitrary complexity.
Concretely, they train agents on exploration of a 3D environment using reinforcement learning and then ask them a set of non-trivial questions. This includes unseen combinations of seen attributes ("zero-shot"), showing that, what the agents learn, is to some degree compositional. Importantly, agents are not trained to answer questions explicitly.

The authors investigate multiple agents and find that LSTM and CPC|A representations are no better than chance, SimCore's representations seem to be the best for the QA task, and there is still a big performance difference between SimCore and the upper bound "No SG".

I think this paper is interesting and well done. I agree with the authors that QA is an intuitive probing tool, which can be used for similar agent analyses in the future.



**Experience Assessment:**

I have read many papers in this area.

**Review Assessment: Checking Correctness Of Derivations And Theory:**

I assessed the sensibility of the derivations and theory.

**Review Assessment: Checking Correctness Of Experiments:**

I assessed the sensibility of the experiments.

**Review Assessment: Thoroughness In Paper Reading:**

I read the paper at least twice and used my best judgement in assessing the paper.

---

> ### Author Response · Authors · 2019-11-07
> **Author response**
>
> We thank the reviewer for their time and feedback. We are encouraged to hear that you found the paper interesting and well done, and our idea of using question-answering to probe an agent’s internal representations generally applicable to future agent analyses!

---

### Official Review · AnonReviewer1 · 2019-10-23
**Official Blind Review #1**

**Rating:** 6

**Review:**

The authors propose a framework to assess to which extent representations built by predictive models such as action-conditional CPC or SimCore contain sufficient information to answer questions about the environment they are trained/test on. The main idea is to train an independent LSTM (a Question-answer decoder) so that given the hidden state of the predictive model and a question about the environment, it is able to answer the question.

The authors give empirical evidence that the representations created by SimCore contain sufficient information for the LSTM to answer questions quite accurately while the representations created by CPC (or a vanilla LSTM) do not contain sufficient information. Based on the experimental results, the authors argue that the information encoded by SimCore contains detailed and compositional information about objects, properties and spatial relations from the physical environment.

The idea is clearly explained and seems sensible, the paper is well written, the execution is competent and the authors provide a sufficient amount of details so that reproducibility should be possible. As a result, I am positive, however, I think it would be best accepted as a workshop paper given that:

- The experiment are only carried out on a single environment, however, their claims are rather general. To support such general claims, experiments on additional environments seem necessary.

- While the idea is sensible, the study is quite narrow because it only compares three models.

- While sensible, the methodological contribution is rather straightforward.

- The take home is quite brief.

**Experience Assessment:**

I have read many papers in this area.

**Review Assessment: Checking Correctness Of Derivations And Theory:**

I assessed the sensibility of the derivations and theory.

**Review Assessment: Checking Correctness Of Experiments:**

I assessed the sensibility of the experiments.

**Review Assessment: Thoroughness In Paper Reading:**

I read the paper at least twice and used my best judgement in assessing the paper.

---

> ### Author Response · Authors · 2019-11-07
> **Author response**
>
> We thank the reviewer for their feedback. We are happy to hear that you found our paper well-written, the experiments thorough, and the experimental settings clearly explained to aid reproducibility. We respond to specific comments below.
>
> 1. Experiments on multiple environments
>
> We agree that experiments across multiple environments would provide stronger empirical evidence. We are setting up a parallel task and experiments in DM-Lab [1] — wherein we train agents with the same exploration reward and evaluate the representations learnt (by an LSTM agent, a CPC|A agent, and a SimCore agent) using a QA decoder on the “color” task. The vocabulary of objects, colors, and visual inputs differ from the environment we reported results on in our submission.
>
> We will follow up with an update as soon as possible once we have these results.
>
> 2. Study is quite narrow because only three models are compared
>
> To our knowledge, CPC|A [2] and SimCore [3] (published in NeurIPS 2019) are the current state-of-the-art in auxiliary predictive objectives; so along with a vanilla LSTM agent, they seemed to be a solid suite of approaches to compare. Having said that, we are happy and curious to analyze other competitive approaches / baselines we may have missed. Please let us know!
>
> 3. Straightforward methodological contribution / brief take home
>
> Our primary contribution is a task-agnostic linguistic decoder to analyze internal representations developed by predictive agents. Prior work has focused on non-linguistic probing networks trained independently for every property — e.g. MLPs for position and orientation, ConvNets for top-down map as in [2,3]. As noted by R2, language provides an intuitive interface and allows for arbitrary levels of complexity. While the decoder itself is operationalized using common architectural primitives (e.g. language-conditioned LSTM), our higher-level idea is novel and we see the architectural simplicity as a positive, low barrier to entry.
>
> [1]: https://github.com/deepmind/lab
> [2]: Neural Predictive Belief Representations, Guo et al., 2018
> [3]: Shaping Belief States with Generative Environment Models for Reinforcement Learning, Gregor et al., NeurIPS 2019

---

> > ### Author Response · Authors · 2019-11-13
> > **Additional results in the DeepMind Lab environment**
> >
> > We set up the “color” task in the DeepMind Lab [1] environment. The environment consists of a rectangular room that is populated with a random selection of objects of different shapes and colors in each episode. There are 6 distinct objects in each room, selected from a pool of 20 objects and 9 different colors. We use a similar exploration reward structure as in the experiments in the main paper to train the agents to navigate and observe all objects. In each episode, we introduce a question of the form `What is the color of the <shape>?' where <shape> is replaced by the name of an object present in the room.
> >
> > Consistent with trends in the main paper, internal representations of the SimCore agent lead to the highest question-answering accuracy, while CPC|A and the vanilla LSTM agent perform worse and similar to each other. Crucially, for running these experiments, we did not change any hyperparameters from the experimental setup in the main paper. This demonstrates that our results are not specific to a single environment and that our approach can be readily applied in a variety of settings. Please see Section A.5 for a plot of question-answering accuracy during training and videos of SimCore agent trajectories here (anonymized):  https://drive.google.com/drive/folders/1itmNlZgDhy6YAwlQxT6LMgr3RDU4ULTh?usp=sharing
> >
> > [1]: DeepMind Lab, Beattie et al., 2016

---

### Official Review · AnonReviewer3 · 2019-10-24
**Official Blind Review #3**

**Rating:** 3

**Review:**


########## Post-Rebuttal Summary ###########
The authors engaged actively in the rebuttal discussion and in the process we were able to concretize the motivation of the submission (as a result in increased my score). However, I think that the submission in its current form lacks experimental analysis of the proposed conditional probes, especially the trade-offs on the reliability of the representation analysis when performed with a conditional probe as well as a clear motivation for the need of a language interface. I can therefore not recommend the submission in its current form for acceptance.
(for detailed discussions + suggested experiments please see rebuttal discussions)
#######################################


## Paper Summary
The paper tackles the problem of analyzing the information captured in the learned representation of a deep neural network. It proposes to replace the commonly used "probing networks" that try to directly infer the information from the learned representation, with a language interface in the form of a QA model which is trained post-hoc without propagating gradients into the learned representation. The authors argue that such a language interface provides a more natural interface for probing the information in a learned representation. The paper shows representation analysis results for the internal representations of agents trained on an exploratory task in a simple, simulated household environment similar to DM Lab. The authors conduct additional analysis on representations learned with auxiliary generative and contrastive objectives.

## Strengths
- the language of the paper is clear and easy to follow
- the paper covers the related work well
- the provided explanations help understand the content of the paper
- the analysis of how the information captured in the agent's representation changes over the course of a trajectory is interesting (more such visualizations in the appendix would be nice!)

## Weaknesses
(1) the motivation for the proposed problem does not convince: why do I need to train a Q/A system to infer which components of the true state are captured by the learned representation? For each of the properties of the environment I could train a probing network (as is done e.g. in [1,2]) and would get much more precise answers; ground truth labels need to be available whether I train QA or probing networks. The authors provide two motivations for the proposed approach which both do not convince me:
	(a) QA "provides an intuitive investigation tool for humans": I cannot imagine a workflow in which researchers would prefer to ask questions to their model over a plot showing explicit regression accuracies aggregated across many data samples (which the conventional probing networks provide).
	(b) "the space of questions is open-ended, [...] enabling comprehensive analysis of [an agent's] internal state": for each new question type we need to provide a sufficiently large number of question-answer-pairs to train the QA system for this question type. we could therefore also train a probing network using the same labels and would get a better overview of whether the state information is captured in the representation. I fail to see how substituting probing networks with a language interface helps the investigation of the representation's properties.

(2) the paper only provides minor novelty: the main proposal is to replace the probing networks, which were extensively used in prior work, with a natural language interface; a proposal that does not seem to provide a clear advantage (see above). The paper does not provide any further technical novelty.

(3) it is possible that the analysis of the contrastive model could improve substantially with a different sampling scheme of the negative examples: maybe sampling from different sequences makes the task of discriminating too easy so that the model is not encouraged to learn a rich representation. It would be interesting to see whether the representation captures more information if a more standard contrastive objective is chosen that discriminates between future frames with different offset in the same sequence (see for example the objective in [3]).

(4) it seems that both environment and chosen task will have high influence on the representation the agent can learn from the collected data. Therefore the fact that the authors evaluate their approach only on a single environment / task combination, both of which they introduce themselves, weakens any conclusions the authors draw from their experiments. It would help strengthen the message of the paper to apply their methodology to previously proposed environment / task combinations, for example in the AI2-THOR environment [4]


[Novelty]: minor
[technical novelty]: minor
[Experimental Design]: not comprehensive
[potential impact]: low


#########################
[overall recommendation]: Reject - In its current form the paper does not provide a convincing argument for why learning a language interface for probing a representation is better than learning the usual probing networks. Further there are some doubts on the setup of the contrastive objective and the paper lacks comprehensive evaluation on standard environments. Therefore I cannot recommend acceptance in its current form.
[Confidence]: High


[1] Neural Predictive Belief Representations, Guo et al., 2018
[2] Shaping Belief States with Generative Environment Models for Reinforcement Learning, Gregor et al., 2019
[3] Representation Learning with Contrastive Predictive Coding, Van den Oord et al., 2018
[4] AI2-THOR: An Interactive 3D Environment for Visual AI, Kolve et al., 2017



#### Final Rebuttal comment (to make it visible to the authors) ####
I understand the authors point that every method for probing representations can potentially be flawed in that the probing mechanism might not be expressive enough to extract the information that is indeed present in the representation. The point I was trying to make in my previous response is, that in the case of conditional probes that try to solve multiple such "probing tasks" in parallel, such uncertainties accumulate because the probing mechanism might trade off performance on one task for performance on another (if solving all of them at once is too challenging). If the submission's key contribution is to make probes conditional this seems like a trade-off that should be experimentally investigated, as it is vital to practitioners how much they can trust their probing method to extract the relevant information if it is actually in the representation.

One possible experiment would be to train an unconditional probe per attribute (maybe on a subset of all possible attribute-object combinations) and then a conditional probe across all of them to show that the conditional probe's assessments of the representation agree with the ensemble of unconditional probes. In my previous response I additionally raised the point that even if it can be shown that conditional probes are reliable, the authors still need to provide arguments why that conditioning should work via a language interface, not for example a symbolic/one-hot interface. If the claim is that this allows for generalization, it should again be shown that this generalization does not increase the error of the probe substantially, such that meaningful conclusions about the representation are still possible.

Regarding testing on more diverse environments: the questions these experiments are supposed to answer (i.e. how reliable are conditional probes) are inherently empirical, so verifying across diverse environments will make the analysis more conclusive.

I agree with the author's point that training 7k unconditional probes would not be practical, probably the current approach would be to train an image decoder that then reconstructs a top-down view of the whole scene. This, however, would have the same problems as a conditional probe, i.e. the probing decoder could trade-off performance between reconstructing different parts of the image. Therefore, I agree with the authors that investigating conditional probes is an interesting direction, but the submission does not provide a comprehensive analysis of this question.

On a final note, I appreciate the continued, factual discussion with the authors and think that the refinement of the focus towards conditional vs unconditional probes is a step in the right direction. To acknowledge that I am raising my score from "reject" to "weak reject". Yet, I think that the work in its current form lacks experimental analysis of the proposed conditional probes. During the rebuttal discussion I highlighted concerns and proposed possible experiments. I cannot recommend acceptance of the submission in its current form, but I encourage the authors to incorporate the refined motivation and add more comprehensive experimental evaluation for a possible resubmission.

**Experience Assessment:**

I have read many papers in this area.

**Review Assessment: Checking Correctness Of Derivations And Theory:**

I carefully checked the derivations and theory.

**Review Assessment: Checking Correctness Of Experiments:**

I carefully checked the experiments.

**Review Assessment: Thoroughness In Paper Reading:**

I read the paper thoroughly.

---

> ### Author Response · Authors · 2019-11-07
> **Author response**
>
> We thank the reviewer for the detailed review! We are encouraged that you found the discussion of related prior work thorough, the analysis of the agent’s representations as it changes over the course of a trajectory interesting (more qualitative videos here (anonymized): https://drive.google.com/drive/folders/1z1oQc-f8IsbsptMqCzYdGeI8qS33zbP2 ), and the writing easy to follow. We understand your concerns and respond to specific comments below.
>
> 1. Motivation for using a language interface to probe emergent semantics
>
> Our motivation for this work is twofold:
> a. How can we understand the emergent knowledge in neural net-based agents as they learn and explore their world?
> b. Is (or when is) an agent’s internal representation sufficient to support propositional knowledge about the environment, compositionality and (eventually) language understanding and use?
>
> With respect to (a), we agree that our work builds on [1,2] (as also noted in Related Work). However, when considered as a general-purpose method for agent analysis, our technique is substantially different from prior work in that our decoder is conditioned by an external input (the question). We use a single, general-purpose network for all question types (e.g. shape, color, etc.), and the question we condition on is provided externally (and has multiple instantiations per question type being processed by the same network, e.g. “What shape is the _blue_ object?”, “What shape is the _red_ object?” for the “shape” question type). This is in contrast to prior work, where 1) there is typically no external input to the probe, and 2) probes have property-specific inductive biases — e.g. MLPs for position and orientation, ConvNets for top-down map as in [1,2].
>
> An additional advantage of having a probe conditioned on external input is that it enables testing for generalization of an agent’s internal representation to perturbations of questions it is trained with. We do this in Sec 5.2, where we hold out some combinations of external input-output pairs (QA pairs) from the training set, e.g. "Q: what shape is the blue object?, A: table” is excluded from the training set of the QA decoder, but “Q: what shape is the blue object?, A: car” and “Q: What shape is the green object?, A: table” are part of the training set (but not the test set).
>
> With respect to (b), we would ultimately like to build an agent that we can interact with in open-ended natural language. While still far from that goal, our work is a step in that direction in that it provides a general-purpose language interface to check if an agent represents facts about the world (e.g. ‘the sofa is red’, or ‘there are four pencils on the floor’). In fact, property-specific probing networks are inherently constrained to the limited set of properties we can enumerate upfront. In future work, we would like to crowdsource open-ended natural language question-answer pairs from humans. In that setting, there is no scalable way to exhaustively enumerate all properties to decode from the agent. And so building independent probes might not even be scalable, while the architecture we propose in this work can be trained as is on that data.
>
> Please let us know if this addresses your concerns. We will include this discussion in the paper.
>
> 2. Results on a different environment / task combination
>
> We agree that results across diverse environments would provide stronger empirical evidence. Unfortunately, no other environment readily provides a set of questions and semantic annotations out-of-the-box; and so we had to set up our own. We are in the process of setting up a parallel task and experiments in DM-Lab [3] — wherein we train agents with the same exploration reward and evaluate the representations learnt (by an LSTM agent, a CPC|A agent, and a SimCore agent) using a QA decoder on the “color” task. The vocabulary of objects, colors, and visual inputs differ from the environment we reported results on in our submission.
>
> We will follow up with an update as soon as possible once we have these results.
>
> 3. CPC with a different negative sampling approach
>
> We experimented with multiple sampling strategies for CPC (whether or not negatives are sampled from the same trajectory, the number of prediction steps, the number of negative examples) and reported the best in the main paper. We have added a more complete discussion in Sec A.1.5.
>
> [1]: Neural Predictive Belief Representations, Guo et al., 2018
> [2]: Shaping Belief States with Generative Environment Models for Reinforcement Learning, Gregor et al., NeurIPS 2019
> [3]: https://github.com/deepmind/lab

---

> > ### Author Response · Authors · 2019-11-13
> > **Additional results in the DeepMind Lab environment**
> >
> > We set up the “color” task in the DeepMind Lab [1] environment. The environment consists of a rectangular room that is populated with a random selection of objects of different shapes and colors in each episode. There are 6 distinct objects in each room, selected from a pool of 20 objects and 9 different colors. We use a similar exploration reward structure as in the experiments in the main paper to train the agents to navigate and observe all objects. In each episode, we introduce a question of the form `What is the color of the <shape>?' where <shape> is replaced by the name of an object present in the room.
> >
> > Consistent with trends in the main paper, internal representations of the SimCore agent lead to the highest question-answering accuracy, while CPC|A and the vanilla LSTM agent perform worse and similar to each other. Crucially, for running these experiments, we did not change any hyperparameters from the experimental setup in the main paper. This demonstrates that our results are not specific to a single environment and that our approach can be readily applied in a variety of settings. Please see Section A.5 for a plot of question-answering accuracy during training and videos of SimCore agent trajectories here (anonymized):  https://drive.google.com/drive/folders/1itmNlZgDhy6YAwlQxT6LMgr3RDU4ULTh?usp=sharing
> >
> > [1]: DeepMind Lab, Beattie et al., 2016

---

> > > ### Comment · AnonReviewer3 · 2019-11-13
> > > **Motivation improved, but not enough evidence to support claims**
> > >
> > > Thanks for the detailed answer to my review!
> > >
> > > Given their reply, the authors seem to agree that the current way of probing the knowledge captured in a network's representations is via training probing networks to infer the information of interest. From what I understand the submission is proposing Question Answering as an alternative tool for investigation.
> > > In their reply the authors provide a somewhat different motivation from the one that was given in the introduction of the submission. Instead of talking about a "more intuitive investigation tool for humans" they now underline the fact that their network is input-conditioned while usual probing networks are not. I think that the paper will benefit from incorporating this clear contrast to probing networks into the introduction; in the current form the introduction does not mention probing networks at all, even though they are the current natural choice for investigating representations.
> > >
> > > I am however still not convinced by the motivation in the light of the experimental evidence that is provided: one good reason to have one probing network per task and not a global one that is input conditioned is, that in such a "multi-task probing network" it would not be clear whether some piece of information is actually not captured in the representation under investigation or whether the probing network traded off performance on this task vs performance on some other task. For question answering systems the problem would be the same: did the network trade off performance for answering questions about the shape for performance on questions about the color or does the representation actually capture shape worse than color?
> > >
> > > Further, even if we assume that training an input conditioned probing network is desirable, it is not clear to me why we would need to train a language interface model vs one that operates for example on symbolic inputs (e.g. one 1-hot vector representing the object type, another 1-hot vector representing the attribute to be inferred). Such a network could still show the generalization capabilities the authors mention but would not require language input.
> > >
> > > In the current form the submission merely shows that it is possible to use QA to infer properties of a representation but does not provide any experimental comparison to prior methods for investigating representations. If, however, the motivation of the work is to propose an alternative investigation tool such comparisons are crucial. The authors should validate that the multi-task structure they introduce does not induce any additional noise in the investigation of the representation's properties (e.g. by training individual probes for each property and showing that the multi-task probe agrees with the single-task ones on which representation is good/bad). Finally, the authors need to provide evidence for why a language input is better suited than a symbolic version.
> > >
> > > To claim that QA is a general tool for testing representations the authors should additionally provide evidence that above-mentioned experimental findings hold across sufficiently different environments. The DMLab experiments the authors provided as part of the rebuttal does not provide such insights as it basically replicates one experiment from the paper with different objects in a new simulator. Instead the authors should provide results on standard environments that are sufficiently different, like AI2-THOR (that I mentioned in my original review) or DMLab for that matter vs Atari (like in [5]). Here the tasks would actually vary (questions about object shape/color in AI2-THOR/DMLab vs questions about score/lives/inventory in Atari).
> > >
> > > Finally, I don't naturally buy the idea of crowdsourcing Q/A pairs for investigating representations that the authors mention in their rebuttal. It is not clear whether such a process would provide sufficient Q/A pairs of a particular type to successfully train a QA model on it or how much the resulting performance deficiencies of the QA model would weaken the statements that can be made about the learned representation. Therefore, if the authors want to make this claim, they need to provide experimental evidence for it.
> > >
> > > ###############
> > > In summary, I appreciate the refined motivation that the authors provide, but I don't see sufficient experimental evidence to support the claims of introducing a better analysis tool, as the manuscript does not compare to any alternative method for investigating learned representations. As such I don't see grounds to increase my score.
> > >
> > >
> > > [5] Unsupervised State Representation Learning in Atari, Anand et al., NeurIPS 2019

---

> > > > ### Author Response · Authors · 2019-11-14
> > > > **Uncovering an agent's propositional knowledge is a useful advance on prior work**
> > > >
> > > > Thanks again for detailed response. We really appreciate it. It's clear you feel strongly that this work does not make a meaningful contribution to the literature. We will try one more attempt to convince you that it is in fact a meaningful advance on prior work and therefore a valuable contribution. Assessing the scale of a contribution is necessarily somewhat subjective, but we feel that a score of 1/10 is entirely incongruous with the amount of progress already represented in this work
> > > >
> > > > Why this work is meaningfully different from prior work
> > > >
> > > > Centuries of research in epistemology are testament to the fact that, as humans, we perceive some distinction between "knowing how" (procedural knowledge) and "knowing what" (propositional knowledge). See e.g. IEP [ https://www.iep.utm.edu/epistemo/ ] for a nice definition of propositional knowledge. It is uncontroversial that deep RL agents can develop a degree of procedural knowledge as they learn to play games, solve tasks etc. However, the extent to which they can develop propositional knowledge is much less clear. Propositional knowledge is important here, firstly because it may ultimately support procedural knowledge in carrying out truly complex tasks, but also because there seems (to philosophers at least) to be something essentially human about having propositional knowledge. If one of the goals of the AI effort is to build learning machines that can engage with, and exhibit convincing intelligence to, human users (.e.g such that humans understand and trust them), then some need for measuring and demonstrating propositional knowledge will arise.
> > > >
> > > > It seems clear, to us at least, that the method we propose in this work provides far greater scope for probing the propositional knowledge of agents than existing approaches. There may be ways to ascertain narrow propositional knowledge from unconditional probes described in prior work, but this would very quickly become contrived, and it would be impossible to measure the breadth or diversity of propositional knowledge encoded by an agent, particularly in a way that would immediately convince human users/observers.
> > > >
> > > > This focus on propositional knowledge is implicit in the current version of the paper, but we shied away from making it explicit for fear of descending unnecessary into the philosophical weeds. However, if you recommend, we would be happy to re-introduce this justification into the paper.
> > > >
> > > > We agree that your suggestion of direct calibration between QA probes to individual property-specific probes (conditional or unconditional) would make an interesting analysis, and also that expansion of the method beyond the two environments that we consider here is an important goal. However, we would contend that not all of these challenges need to be met immediately, and that the introduction and proof of concept of the probing method in this paper is a sufficient contribution on its own and paves the way for additional work in this direction.

---

> > > > > ### Comment · AnonReviewer3 · 2019-11-15
> > > > > **Further analysis required to convince that this is a useful advance on prior work**
> > > > >
> > > > > Thanks again for the quick response!
> > > > > I do not think that it is necessary to include the focus on propositional knowledge in the intro, I think it is clear what kind of information both probing networks as well as the QA approach are trying to analyze a representation for.
> > > > >
> > > > > In their response the authors claim that "the method [they] propose in this work provides far greater scope for probing the propositional knowledge of agents than existing approaches". However, I repeat the statement from my last response: the current submission lacks experimental evidence for such claims. While QA might provide a simple interface for training one model to elicit information about a multitude of attributes, it is unclear whether that is actually desirable and what tradeoffs go along with that. In my last response I highlighted the problem that the model can tradeoff performance between tasks in such a multi-task setup, so that we don't know whether the representation actually does not capture the information or whether it is too hard of a question for the QA system to answer in the first place / whether the QA model decides to focus on other questions. In Tab.2 of the submission the authors report "oracle" QA performance of as low as 41% on some question types (with representations optimized for that task), indicating that the QA task itself is hard and that the concern about the noise introduced by the interface is warranted. Experiments that compare to non-conditional baselines across multiple environments are needed to support the author's claims.
> > > > >
> > > > > In conclusion, the submission proposes the interesting idea to investigate representations via question answering and shows that it is possible to train a QA system on an unsupervised representation. It does not draw comparisons to prior methods for analyzing representations, it does not analyze the tradeoffs of input-conditioned probing interfaces and it does not provide sufficient evidence that the QA score is indicative of the information captured in the representation. I therefore do not recommend acceptance of the submission in its current form, but we detailed both sides' arguments in this thread and it is now upon the AC to make a decision.
> > > > >
> > > > >
> > > > > P.S.: Regarding the scoring scale: please note that this year's ICLR scoring scale only contains four score options [1, 3, 6, 8]. In last year's scoring system I would see my score rather like a 3, indicating that I don't think that the submission in its current form is ready for publication, but that there is an idea that can lead to a successful publication. In that sense I encourage the authors to resubmit a revised version of the manuscript to one of the upcoming conferences!

---

> > > > > > ### Author Response · Authors · 2019-11-15
> > > > > > **We can't see how the proposed remedies would resolve your principal concern**
> > > > > >
> > > > > > Thank you for your response. We really appreciate your continued engagement, which has been instrumental in the improvements to the paper that we have made so far.
> > > > > >
> > > > > > So that we can understand exactly how to address your worries, we just want to dig a little deeper into your principal concern, namely, that our methods cannot definitively determine if information exists in representations or whether it's just being obscured because of the weakness of the QA decoder ('[our method] does not provide sufficient evidence that the QA score is indicative of the information captured in the representation'). Is it not the case that this flaw is inherent to any probing approach, including all of the prior work that you refer to in your comments? Any failure of a probing analysis to identify information may be due to insufficient power or expressivity of the decoder. We acknowledge this in the paper, which is why we explicitly measure accuracy as a function of decoder capacity. Your position seems to be that this flaw is terminal for a QA conditional decoder, but acceptable for a non-conditional task-specific probe.
> > > > > >
> > > > > > You suggest several ways that our paper might be improved (thank you), but it's really unclear to us how taking these steps would resolve your concern. First, you recommend comparing our decoder to an unconditional probe. As explained above, neither unconditional nor conditional probes are guaranteed to uncover *all* the information in a representation. Employing unconditional probes in a way that spanned the same range of propositional knowledge as we consider in the paper would require us to train, by our estimates, 7660 independent probing networks (what is the colour of the table? What is the colour of the chair? Etc etc). This would involve a vast effort, but having done so, we would still not know definitively what information is in the agent's representations. At some point, the practicality of executing the method must be a relevant factor in assessing its merit. Given that probing methods cannot (whether conditional or unconditional) reveal the extent of knowledge in a representation, we find the more pragmatic question of how likely are they to be used to uncover or reveal the knowledge of an agent to be more compelling.
> > > > > >
> > > > > > We believe there is a very real chance that, without much additional effort, researchers might choose to bolt on a conditional (QA-style) decoder in order to monitor the aggregation of knowledge in their agents. Doing so with thousands of independent unconditional decoders seems highly impractical. We understand that you are strongly opposed to this position, but to help us move forward with this work, what would you suggest is a more likely, or less flawed, way that users will keep track of the propositional knowledge acquired by agents?
> > > > > >
> > > > > > You have also requested that we run our method on more than two environments. Again, before embarking on this we face the same question. How would this resolve the fundamental criticisms of the method that are leading you to recommend rejection of the paper here? It would certainly be indicative of how the method is easy and quick to apply (although we think that our quick and dirty experiment with DM-lab and no tuning serves this purpose), but ultimately, exactly the same criticisms could easily be leveled at the work. Why would three environments suddenly be sufficient to overcome these issues?
> > > > > >
> > > > > > Unfortunately, we have the same uncertainty around your request for us to "draw comparisons to prior methods for analyzing representations" (without a 'right' answer to the question of what information *should* exist in an agent, how exactly might we compare two methods?), and for "an analysis of the tradeoffs of input-conditioned probing interfaces" (the 'tradeoffs' seem to involve the pragmatic and philosophical issues that we have discussed in this review process - how might we add that satisfactorily to the paper?).
> > > > > >
> > > > > > Thanks again for your hard work and thoughtful assessment of our paper. We are genuinely grateful for it.

---

### Decision · Program_Chairs · 2019-12-19

**Decision:**

Reject

**Comment:**

This paper proposes question-answering as a general paradigm to decode and understand the representations that agents develop, with application to two recent approaches to predictive modeling. During rebuttal, some critical issues still exist, e.g., as Reviewer#3 pointed out, the submission in its current form lacks experimental analysis of the proposed conditional probes, especially the trade-offs on the reliability of the representation analysis when performed with a conditional probe as well as a clear motivation for the need of a language interface. The authors are encouraged to incorporate the refined motivation and add more comprehensive experimental evaluation for a possible resubmission.